# Feasibility and preliminary efficacy of heavy Lifting Strength Training versus Usual Care in Head and Neck Cancer Survivors (the LIFTING 2 Trial): A study protocol for a single-centre, phase II, randomized controlled trial

**Stephanie M. Ntoukas**[1], **Margaret L. McNeely**[2], **Carla M. Prado**[3], **Kerry S. Courneya**[1]*

**1** Faculty of Kinesiology, Sport, and Recreation, College of Health Sciences, University of Alberta, Edmonton, Alberta, Canada, **2** Faculty of Rehabilitation Medicine, College of Health Sciences, University of Alberta, Edmonton, Alberta, Canada, **3** Department of Agricultural, Life and Environmental Sciences, College of Natural and Applied Sciences, Edmonton, Alberta, Canada

* kerrycourneya@ualberta.ca

## Abstract

### Background

Despite improvements in treatments, head and neck cancer survivors (HNCS) still endure acute and chronic side effects such as loss of muscular strength, limitations in physical function, fatigue, and swallowing difficulties that impact quality of life (QoL). Light-to-moderate intensity strength training (LMST) has been shown to improve some of these side effects. Heavy lifting strength training (HLST) may further improve outcomes, however, only one pilot study has focused on HNCS. The primary aim of this study is to further establish the feasibility of HLST in HNCS. A secondary aim is to provide preliminary evidence of the effects of a HLST program compared to Usual Care (UC) in HNCS.

### Methods

This single-centre, two-armed, randomized controlled trial will aim to recruit 48 HNCS ≥1-year post-treatment, and randomly assign them to a 12-week Heavy Lifting Strength Training (HLST) group or Usual Care (UC) group. With 80% power, a two-tailed alpha of $p < 0.05$, and adjusting for covariates that explain 25% of the variance in the outcome, we will be able to detect a standardized effect size of 0.80 with 38 (19/group) evaluable HNCS, allowing for a 20% loss to follow-up. The HLST group will exercise twice weekly, progressing to lifting low repetitions of heavy loads at 80% to 100% of maximal perceived exertion, whereas the UC group will not receive any exercise prescription or instruction during the 12-week intervention. The primary efficacy outcome will be upper and lower body muscular strength assessed by 3RM

**Data availability statement:** Deidentified research data will be made publicly available when the study is completed and published.

**Funding:** Dr. Kerry Courneya will be fully funding this trial out of his Canadian Institutes of Health Research Foundation Grant and his Canada Research Chair funding. This will cover costs of study personnel including casual employees and research assistantships for graduate students, and the cost to conduct bioelectrical impedance analyses for the body composition assessment of all study participants, at baseline and post intervention. Dr. Courneya's funding will also cover the costs of incidentals such as printing letters and study brochures to mail out for participant recruitment, cleaning supplies for the laboratory, and transportation to and from our facility for all study participants. Dr. Carla Prado was partially funded through Canada's Research Chair program.

**Competing interests:** The authors have declared that no competing interests exist.

**Provenance:** Not commissioned; externally peer-reviewed.

tests. Secondary efficacy outcomes will include health-related fitness and patient-reported outcomes assessed with reliable physical assessments and validated questionnaires.

## Significance

Our findings will answer the important question of whether lifting heavier weights is feasible for HNCS and leads to better outcomes compared to lifting no weights. The results will inform future studies comparing exercise intensities in HNCS and other cancer populations.

---

## Introduction

Head and neck cancers (HNCs) are complex and diverse tumours, originating in the oral cavity, oropharynx, hypopharynx, nasopharynx, lip, larynx, paranasal sinus, salivary gland, and mucosal melanoma [1]. HNCs are in the top 10 most common cancers worldwide, and make up approximately 5% to 7% of solid tumours globally [2–6]. Standard treatment for early stage HNC is surgery or radiotherapy. Multiple modalities, mainly chemoradiotherapy, are used for locally advanced HNC [7]. Despite improvements in treatments, head and neck cancer survivors (HNCS) still endure numerous acute and chronic side effects including: dental and oral complications, nutritional, speech and voice impairments, immune suppression, infectious complications, shoulder dysfunction, pain, shortness of breath, reduced muscle strength and mass, physical fatigue, difficulty sleeping, swallowing dysfunction and affected appetite, self-consciousness, embarrassment unattractiveness, low self-esteem, and reduced quality of life (QoL) [7–15].

Strength training improves some of these side effects in HNCS [16–22], however, most studies to date have tested light to moderate strength training (LMST) rather than training with heavy loads [14,16,17,20,22–25]. LMST involves lifting lighter loads more times (i.e., 10–15 repetitions) whereas heavy lifting strength training (HLST) involves lifting heavier loads fewer times (i.e., 1–6 repetitions), typically, 80% to 100% of one repetition maximum (1RM) [26]. While LMST interventions have been shown to be beneficial for HNCS, preliminary HLST research in HNCS and other populations have shown that training with heavy loads may be feasible and necessary in order to maximize muscular strength gains [26–28]. These findings are promising and raise the question on whether or not HLST may lead to greater improvements in skeletal muscle mass, strength, and function, and QoL in HNCS.

Here, we propose the Feasibility and Preliminary Efficacy of Heavy Lifting Strength Training versus Usual Care in Head and Neck Cancer Survivors (the LIFTING 2) trial, the first phase II, randomized controlled trial that will further establish the feasibility of HLST using free weights, and provide preliminary efficacy data compared to Usual Care (UC) in HNCS. The primary efficacy outcome is upper and lower body muscular strength changes from baseline to post-intervention, assessed by reliable 3 repetition maximum (3RM) tests. Secondary efficacy outcomes include physical

function, shoulder mobility, and handgrip strength assessed by validated physical assessments; body composition components such as skeletal muscle mass and body fat percentage assessed by bioelectrical impedance; and quality of life (QoL), fear of cancer recurrence, symptom burden, fatigue, self-esteem, sleep, swallowing abilities, and nutritional status assessed by well-validated questionnaires. We hypothesize that HLST will be safe and feasible for HNCS, and anticipate no adverse events to occur which are directly related to HLST [28–30]. Lasty, we expect that HLST will be superior to UC for muscle strength, mass, and function, and QoL.

## Methods

### Study design

The LIFTING 2 trial will be conducted as a single-centre, two-armed, phase II, randomized controlled trial at the University of Alberta in Edmonton, Alberta, Canada. Participants will be randomly assigned to the 12-week HLST or UC group. Participants will be stratified based on sex and cancer stage (I-III vs. IV). The proposed flow of participants in the LIFTING 2 trial is summarized in **Fig 1**.

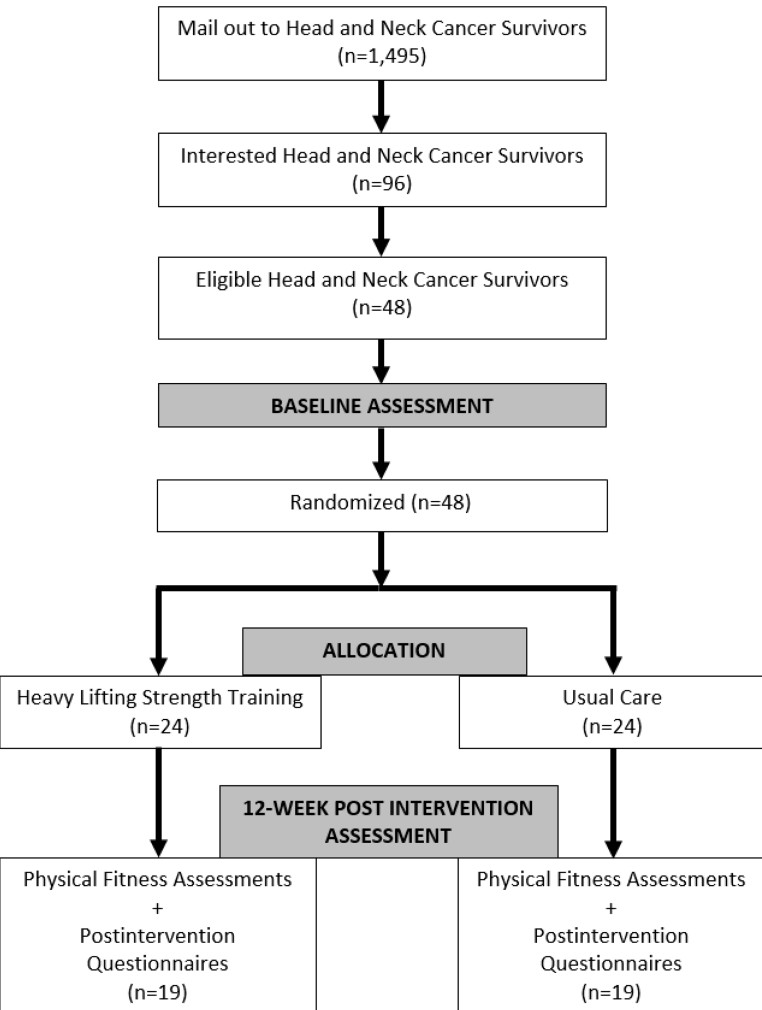

**Fig 1. CONSORT flow diagram.**

---

## Study population

Eligible participants will include males and females who meet the following criteria: 1) any HNC subtype, cancer stage (I-IV), and treatment type (i.e., surgery, chemotherapy, radiation, combination treatment); 2) at least one-year post-treatment of any kind with adequate shoulder range of motion (active flexion and abduction) or recovery of the spinal accessory nerve (SAN); 3) 18 years and older; 4) no unmanaged medical conditions; 5) approved for exercise by oncologist and a certified exercise physiologist or kinesiologist; 6) ability to understand and communicate in English; 7) has not met the Canadian Physical Activity strength training guidelines within the past one-month: *at least two days per week of muscle and bone strength training activities using major muscle groups*. and 8) participants will be excluded if they are currently involved in a different exercise trial or clinical drug trial. Our target accrual number is 48 patients over 12-months. Participant recruitment and data collection will begin in March 2025 and end in March 2026. Results will be collected on an ongoing basis during this timeframe, but will be analyzed and interpreted between April and June 2026. See **Table 1** for a schedule of enrollment, interventions, and assessments.

## Recruitment and screening

Potentially eligible participants will be identified and contacted via the Alberta Cancer Registry, a database that holds information pertaining to all living and deceased individuals diagnosed with cancer in the province of Alberta, Canada. By recruiting in this way, we will have contacted all potentially eligible HNCS at least one-year post-treatment.

Interested HNCS will contact and meet with the study coordinator (SMN) via an online pre-screening meeting, which will serve as a chance to ask about medical history such as type of cancer, cancer stage, and treatment type, elaborate further on study details, and obtain verbal informed consent at this time. Once online questionnaires are complete in the Research Electronic Data Capture (REDCap), and the study coordinator will schedule interested HNCS for in-person baseline physical assessments at the Exercise Oncology Research Laboratory at the University of Alberta. Finally, eligibility will be determined through objective measurements of active shoulder flexion and abduction ranges of motion. To satisfy eligibility criteria #2 above, participants will be required to meet or exceed the following age-based cut points for shoulder range of motion: participants 18–50 years old: ≥ 150° for flexion and abduction; participants over 50 years old: ≥ 130° for flexion and abduction.

**Table 1. Timeline of enrollment, interventions, and assessments for the LIFTING 2 Trial.**

| | STUDY PERIOD | | | | End of Study |
|---|---|---|---|---|---|
| | Enrollment | Allocation | Post-Allocation | | |
| TIMEPOINT | Baseline | 0 weeks | 6 weeks | 12 weeks | Wrap-Up |
| **ENROLLMENT** | | | | | |
| Eligibility Screening | X | | | | |
| Informed Consent | X | | | | |
| Questionnaires | X | | | | |
| Baseline Assessment | X | | | | |
| Randomization | | X | | | |
| **INTERVENTIONS** | | | | | |
| Heavy Lifting Strength Training | | ←————————————————→ | | | |
| Usual Care | | ←————————————————→ | | | |
| **ASSESSMENTS** | X | | | | X |

## Randomization and blinding

After completing baseline assessments, participants will be randomly assigned to either the HLST Group or the UC Group in a 1:1: ratio using a computer-generated program within REDCap, with random blocks of 4 or 6. The allocation sequence will be generated by and independent researcher and uploaded directly to REDCap to ensure concealment from study personnel. Participants will be stratified by biological sex and cancer stage (I-III vs. IV) to ensure balanced group distribution. Additional stratification variables will be considered in a future phase III trial. The UC Group was chosen as the comparator in order to evaluate the feasibility and efficacy of a HLST program. During the conception of this study, we considered comparing HLST to LMST but felt that we needed to further establish the feasibility of HLST before comparing it to LMST. A future phase III study would then compare HLST to LMST to determine the optimal exercise intensity for HNCS.

Group allocation will be shared with the participant immediately after the baseline assessment. Participants and investigators will not be blinded to group assignment, given the nature of the intervention. Outcome assessors will also not be blinded to group assignment because of logistical challenges. However, we will follow a standardized, detailed protocol and all assessors will be trained on the importance of standardizing outcome assessments and prioritizing safety.

## Maximal strength testing

Two separate 3RM tests will be used for baseline and post-intervention physical assessments to evaluate upper and lower body strength with the leg press and chest press machines. Maximal testing is safe, reliable, and the gold standard for assessing muscle strength in healthy, older adults [31]. Prior to attempting these movements, the study coordinator will give an in-depth explanation of the purpose of the tests, how to safely execute the lifts, and a demonstration.

To begin, after a 5-minute aerobic warm up and standardized dynamic stretches, study participants will perform body-weight and resistance band movements to practice appropriate technique. Progressions leading up to the 3RM are as follows. For the leg press, participants will initially perform 8–10 reps at less than 50% of body weight in pounds. To follow, the reps will decrease as the weight gradually increases based on body weight in pounds: 50% body weight for 5–6 reps, 70% for 2–3 reps, and 85%, 90%, 95%, and 100%+ for 3 repetitions. For the chest press, after warming up with a resistance band, participants will perform 8–10 reps at less than 35% of body weight in pounds. Following this, the reps will also decrease as the weight gradually increases based on body weight in pounds: 35% body weight for 5–6 reps, and 40%+ for 3 reps. 3RM tests will be performed in this order: leg press, followed by chest press.

In order to assess a true 3RM, the following factors will be considered. Since the Borg RPE Scales are primarily used to assess intensity of aerobic exercise and may be inaccurate and difficult to conceptualize for beginner lifters [32], a repetition in reserve (RIR)-based RPE scale will be used to assess exercise intensity, which may be more valid to assess maximal or near maximal strength than RPE alone [32]. After each set during the physical assessments, participants will be asked for their RPE, which corresponded to a number on the RIR scale. Participants will be asked the following question, "If I didn't stop you, how many more repetitions could you have done after the prescribed number?" Conversions from RPE to RIR are as follows: RPE 10 = RIR 0 (maximum effort), RPE 9 = RIR 1, RPE 8 = RIR 2…RPE 1 = RIR 9+ (little to no effort) (44). For maximal testing, participants should report an RIR of 0–1, or RPE of 9–10, in order for the lift to be considered a true maximal test.

In addition to the RIR-based RPE reports, the study coordinator will assess 3RM tests via observation. As the weight increases and the reps decrease while approaching a 3RM, time under tension (total duration of lift) should be progressively longer. Participants should be somewhat struggling to move the weight when a true 3RM has been reached, and should not be able to quickly move the load. If the latter was the case and participants do not want to continue for whatever reason, the reps and load will be plugged into a 3RM prediction equation, and this maximal testing limitation will be recorded.

Finally, attention will be paid to form for both 3RM tests. Participants should maintain the same leg press squat depth even as loads increase, although there is a tendency to shorten range with increasing weight, making the lift easier and shorter in duration (less time under tension). To avoid this issue and maintain consistency in squat depth across all participants, the number at the side of the leg press machine (indicating its depth) will vary between participants and be recorded. This depth will remain the same for baseline and post-intervention assessments for each participant. Lifters will begin in a deep enough position so that the knee joint forms a 90-degree angle. From here, they will push and the starting position will be in the extended position. Lifters will be instructed to squat down by hinging at the hips and bending the knees, until the weight plates come close to each other, then push back up. If the leg press range is shortened with increasing weight, the lifter will be reminded to bring weight plates closer by sinking down deeper with a greater bend in the knees and hip hinge. They will be stopped if the same form and depth cannot be maintained with increasing load.

For the chest press assessment, the seat height for each participant will vary and be recorded. This height will remain the same for baseline and post-intervention assessments to maintain consistency. Lifters will hold onto the vertical handles and push, starting in an extended position. With a neutral wrist, they will then be instructed to lower the weight by bending at the elbows, until the weight plates come close to each other and upper arm is in line with the torso's midline, then push back up. If the chest press range is shortened with increasing weight, the lifter will be reminded to bring weight plates closer to each other by extending at the shoulder, allowing elbows to sink further back. They will be stopped if the same form cannot be maintained with increasing load.

### Intervention

Participants that are randomized to the HLST group will take part in a 12-week, supervised exercise program at a frequency of 2 days per week. The HLST group will progress to lifting heavy loads (80% to 100% of maximal perceived exertion) of low repetitions (i.e., 1–6) for the barbell squat, bench press, and deadlift. If they are not able to safely execute the prespecified strength movements after multiple attempts, the following substitutions were planned for the duration of the 12-week intervention: leg press to replace the barbell squat, dumbbell bench press to replace the barbell bench press, rack pulls or dumbbell deadlifts to replace the barbell deadlift from the floor. Accessory exercises, which target smaller muscles, will also be included in the exercise protocol and include: face pulls, seated rows, dumbbell lunges, farmers carry, and planks. The planned intensity progression of the three primary exercises in the LIFTING 2 trial are presented in **Fig 2**.

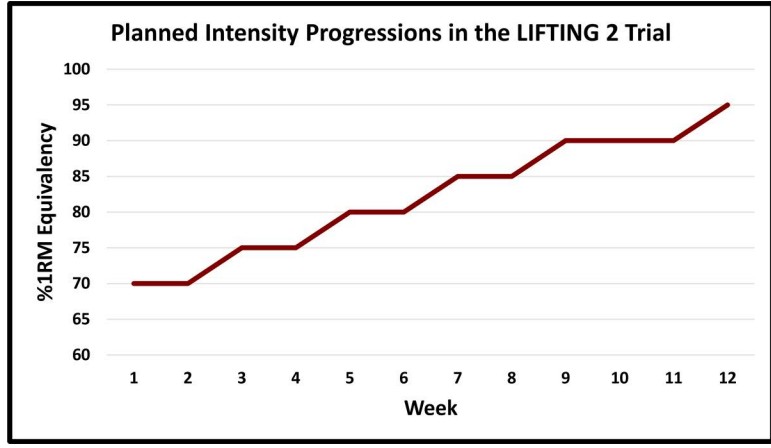

**Fig 2. Planned intensity progressions in the LIFTING 2 trial.** %1RM: percent of one-repetition maximum.

Details of the exercise intensity progression phases are as follows:

In weeks 1–2, the weight is chosen based on warmup sets that would only allow for 10 repetitions (10RM), which is approximately equivalent to 70% to 75% 1RM. In weeks 3–5, the weight is chosen based on warmup sets and previous sessions that would only allow for 8 repetitions (8RM), which is approximately equivalent to 75% to 80% 1RM. In weeks 6–8, the weight is chosen based on warmup sets and previous sessions that would only allow for 5 repetitions (5RM), which is approximately equivalent to 80% to 85% 1RM. In weeks 9–12, the weight is chosen based on warmup sets and previous sessions that would only allow for 1–3 repetitions (≤ 3RM), which is approximately equivalent to 90% to 100% 1RM.

Prior to exercise testing and each exercise session, resting heart rate and resting blood pressure values will be assessed. Additionally, participants will be asked to rate their current level of perceived fatigue and pain, on a scale of 0 (no fatigue/pain) to 10 (worst possible fatigue/pain). All participants in the HLST group will be guided through a 10-minute warm up. This will consist of low intensity biking or walking on the treadmill for 5 minutes, and structured dynamic move-ments such as leg swings, arm circles, hip, shoulder, and ankle mobility exercises. At the end of each session, all par-ticipants will be guided through a 10-minute cool down. This will consist of structured static stretching. Exercise intensity will be gradually progressed over a period of at least 5-weeks to reach heavy loads (80% to 100% of maximal perceived exertion) for the barbell squat, bench press, and deadlift (Table 2).

Participants that are randomized to the UC group will be asked to continue with their typical daily routine and habit of exercise during the 12-week intervention period. They will not receive any information or education regarding exercise. After the post-intervention assessments at 12-weeks, the UC group will be offered a 4-week introduction to HLST program and/or referred to a community-based exercise program.

## Feasibility and safety measures

The feasibility and safety outcomes will be determined based on the interest rate (number of HNCS who contact us), eligibility rate (with reasons for ineligibility), recruitment rate (with reasons for refusal), 3RM testing rate (with reasons for not completing the test), HLST program adherence (including attendance, dose modifications, and progression), follow-up assessment rate (with reasons for drop out), and adverse events.

The following thresholds have been adopted as indicating issues with feasibility requiring protocol modification prior to further efficacy testing: 1) Safety Concerns: > 10% of participants in the HLST group report injuries and/or adverse events likely related to the exercise intervention, which will be reported to our ethics board and the Principal Investigator (KSC) immediately; 2) Recruitment: failure to recruit 48 patients from Edmonton, Alberta which may indicate the need for a multi-centre trial moving forward; 3) Attrition Rate: > 20% of randomized participants drop out of the study; or 4) Adher-ence: 80% of participants adhere to the protocol.

## Prespecified criteria for stopping or changing the study protocol due to safety concerns

A participant may be discontinued at any time if the participant, the investigator, or the primary physician feels that it is not in the patient's best interest to participate in the study, the physical assessments, and/or questionnaires. This includes any

**Table 2. The 12-week, heavy lifting strength training periodization scheme in the LIFTING 2 trial. 1RM: one repetition maximum.**

| Week | 1 | | 2 | | 3 | | 4 | | 5 | | 6 | | 7 | | 8 | | 9 | | 10 | | 11 | | 12 | |
|---|---|---|---|---|---|---|---|---|---|---|---|---|---|---|---|---|---|---|---|---|---|---|---|---|
| Exercise Sessions | 1 | 2 | 3 | 4 | 5 | 6 | 7 | 8 | 9 | 10 | 11 | 12 | 13 | 14 | 15 | 16 | 17 | 18 | 19 | 20 | 21 | 22 | 23 | 24 |
| Repetitions | 10 | | 10 | | 8 | | 8 | | 8 | | 5 | | 5 | | 5 | | 1-3 | | 1-3 | | 1-3 | | 1-3 | |
| 1RM Equivalency (%) | 70-75% 1RM | | 70-75% 1RM | | 75-80% 1RM | | 75-80% 1RM | | 75-80% 1RM | | 80-85% 1RM | | 80-85% 1RM | | 80-85% 1RM | | 90-100% 1RM | | 90-100% 1RM | | 90-100% 1RM | | 90-100% 1RM | |

adverse events (related or unrelated to the study intervention) that cannot be ameliorated by the use of adequate medical intervention, exercise intervention modifications, or would lead to undue risk if the participant were to continue with the study procedures.

A resting heart rate ≥100 bpm, resting blood pressure ≥160/90 mmHg will result in no exercise testing or exercise sessions to be conducted for safety reasons. An injury related or unrelated to the exercise intervention will result in program modifications including reducing the load, sets, and/or repetitions, or completely removing an exercise if necessary, depending on the severity of the injury. Exercise intensity will gradually progress overtime. However, volume will be modified or remain unchanged if participants are unable to safely progress. During 3RM testing and exercise sessions, technique and perceived pain will regularly be monitored.

### Outcomes

Participant characteristics, behavioural outcomes, feasibility outcomes, health-related fitness and patient-reported outcomes, physical function, anthropometry and body composition will be assessed in this study. The instruments that will be used and the timepoints of assessments are shown in **Table 3**.

**Demographic variables.** Demographic variables will be assessed using self-report. Medical data will be obtained via self-report by the participants, and include: type of HNC, type of treatment modality and dissection type, cancer recurrences if any, comorbidities, current and previous injuries, and a list of medications.

**Behavioural outcomes.** Social cognitive questionnaires will be informed by the Theory of Planned Behaviour [33]. Alcohol consumption and smoking status will be assessed using self-report. Exercise levels will be assessed via the Godin Leisure Time Exercise Questionnaire [34].

**Feasibility and safety outcomes.** The feasibility and safety outcomes will be determined based on the interest rate (number of HNCS who contact us), eligibility rate (with reasons for ineligibility), recruitment rate (with reasons for refusal), 3RM testing rate (with reasons for not completing the test), HLST program adherence (including attendance, dose modifications, and progression), follow-up assessment rate (with reasons for drop out), and adverse events.

**Health-related fitness outcomes.** 3RM assessments will be used to assess upper and lower body strength with the leg press and chest press machines. RIR will be used to assess the perceived intensity of these exercises and this may be the optimal way to assess exercise intensity in terms of safety, reliability, validity, and tolerability [31]. The 3RM test is reliable and the gold standard assessment to evaluate maximal strength of resistance exercises in 'healthy' men and women, and it can be used by athletic trainers, health and fitness professionals and rehabilitation specialists to quantify the level of strength, to assess strength imbalances, and to evaluate training programs [31,35]. 3RM, RIR, and RPE scales are equally effective at improving muscular strength and functional performance in an older population [31], and we expect the majority of participants to be of middle to older age due to the typical age of onset of HNC.

**Patient-reported outcomes.** Patient-reported outcomes that will be assessed include the following. QoL will be measured using the European Organization for Research and Treatment of Cancer QoL Questionnaire-C30 (EORTC QLQ-C30) [36]. Fear of cancer recurrence will be measured using the Fear of Cancer Recurrence Inventory (FCRI) [37]. Specific head and neck cancer treatment symptom burden will be assessed using the Neck Dissection Impairment Index (NDII) [38]. Self-esteem will be measured using the Rosenberg Self-Esteem Scale (RSE) [39]. Typical sleep habits will be measured using the Insomnia Severity Index (ISI) [40]. Swallowing abilities will be assessed using the MD Anderson Dysphagia Inventory [41].

**Physical function.** Physical function assessments for participants 50 years old and up will include: 6 Minute Walk Test (6MWT), and 30 second sit to stand to assess aerobic functional capacity and lower body muscular endurance [42,43]. Active shoulder flexion and abduction ranges of motion will be measured using a goniometer to ensure ranges meet the cut points required for study eligibility outlined above. Handgrip strength will be assessed using a handgrip dynamometer [44].

**Table 3. Outcome measures and assessment timepoints of the LIFTING 2 trial.**

| PARTICIPANT CHARACTERISTICS | INSTRUMENT | BASELINE | POST-INTERVENTION (3 months) |
|---|---|---|---|
| Demographic Variables | Self-Report | X | |
| Medical Information | Self-Report | X | |
| **BEHAVIOURAL OUTCOMES** | **INSTRUMENT** | **BASELINE** | **POST-INTERVENTION (3 months)** |
| Social Cognitive Behaviours (i.e., motivation) | Questions informed by the Theory of Planned Behaviour | X | X |
| Alcohol Consumption | Self-Report | X | |
| Smoking Status | Self-Report | X | |
| Exercise Levels | Godin Leisure Time Exercise Questionnaire (Self-Report) | X | X |
| **FEASBILITY OUTCOMES** | **INSTRUMENT** | **ASSESSED ON AN ONGOING BASIS** | |
| Interest Rate | # (%) HNCS who contact us | X | |
| Eligibility Rate | # (%) HNCS eligible and ineligible with reasons for ineligibility | X | |
| Recruitment Rate | # (%) HNCS randomized with reasons for refusal | X | |
| 3RM Testing Rate HLST Program Adherence | # (%) of exercise sessions completed including attendance, dose modifications, and progression | X | |
| | | **BASELINE** | **POST-INTERVENTION (3 months)** |
| Follow-Up Assessment Rate | # (%) HNCS who completed post-intervention physical assessments and questionnaires with reasons for dropout | | X |
| Adverse Events | # (%) injuries and/or undesired effects related to exercise intervention | | X |
| **HEALTH-RELATED FITNESS OUTCOMES** | **INSTRUMENT** | **BASLEINE** | **POST-INTERVENTION (3 months)** |
| Muscular Strength | | | |
| Lower Body Strength | 3RM leg press | X | X |
| Upper Body Strength | 3RM chest press | X | X |
| **PATIENT-REPORTED OUTCOMES** | **INSTRUMENT** | **BASELINE** | **POST-INTERVENTION (3 months)** |
| Quality of Life | EORTC QLQ-C30 | X | X |
| Fear of Cancer Recurrence | Fear of Cancer Recurrence Inventory | X | X |
| Head & Neck Cancer Symptom Burden | Neck Dissection Impairment Index | X | X |
| Self-Esteem | Rosenberg Self-Esteem Scale | X | X |
| Sleep Quality | Insomnia Severity Index | X | X |
| Swallowing Abilities | MD Anderson Dysphagia Inventory | X | X |
| **PHYSICAL FUNCTION** | **INSTRUMENT** | **BASELINE** | **POST-INTERVENTION (3 MONTHS)** |
| Aerobic Functional Capacity | 6 Minute Walk Test | X | X |
| Lower Body Muscular Endurance | 30 Second Sit to Stand | X | X |
| Shoulder Ranges of Motion | Active Flexion | X | X |
| | Active Abduction | X | X |

*(Continued)*

 

**Table 3.** (Continued)

| PARTICIPANT CHARACTERISTICS | INSTRUMENT | BASELINE | POST-INTERVENTION (3 months) |
|---|---|---|---|
| Handgrip Strength | Handgrip dynamometer (Jamar) | X | X |
| ANTHROPOMETRY AND BODY COMPOSITION | INSTRUMENT | BASELINE | POST-INTERVENTION (3 MONTHS) |
| Fat Mass | Bioelectrical Impedance Analysis (InBody 770) | X | X |
| Lean Compartments | | | |
| Phase Angle | | | |
| Height | Stadiometer | X | X |
| Weight | Beam Scale | X | X |
| Body Mass Index | Weight (kg)/Height (m²) | X | X |
| Waist to Hip Ratio | Measuring tape | X | X |
| Nutritional Status | GLIM and PG-SGA | X | X |

3RM: 3 repetition maximum; EORTC QLQ-C30: European Organization for Research and Treatment Quality of Life Questionnaire Core 30-item; GLIM: Global Leadership Initiative on Malnutrition; PG-SGA: Scored Patient Generated Subjective Global Assessment

**Anthropometry and body composition.** Anthropometry and body composition will be assessed using hip to waist ratio, height, weight using a measuring tape, stadiometer, and a beam scale, respectively. Body mass index (BMI) will also be calculated and categorized based on the World Health Organization BMI classification percentiles [45]. Body composition components such as skeletal muscle mass and body fat percentage, will be estimated using a bioelectrical impedance analysis (BIA) device, the InBody770. BIA is a clinically accessible tool for estimating fat mass, lean compartments, and it also provides a measurements of phase angle. Despite its limitations, it remains valuable for assessing changes over time and has been previously utilized in patients with HNC [46–49]. Body composition will be interpreted alone and in the context of the GLIM phenotypic criteria [50].

Nutritional status will be assessed as a descriptive variable at baseline (within 1-week prior to starting the HLST program or UC period) and at post-intervention (within 1-week of completing the HLST program or UC period). Two primary tools will be used: the Global Leadership Initiative on Malnutrition (GLIM) and the Scored Patient Generated Subjective Global Assessment (PG-SGA) [50,51]. Specifically, the PG-SGA will supplement the evaluation to fulfill the criteria set by GLIM. If malnutrition is suspected, the research team will advise the study participant to seek appropriate professional care. However, this is not an exclusion criterion, and individuals suspected of having malnutrition may still participate in the study if they meet eligible criteria and wish to enroll.

## Sample size

Our goal will be to recruit 48 HNCS. With 80% power, a two-tailed alpha of $p < 0.05$, and adjusting for covariates that explain 25% of the variance in the outcome, we will be able to detect a standardized effect size 0.80 with 38 (19/group) evaluable HNCS allowing for a 20% loss to follow-up. We believe that this large effect on strength is realistic in our study, given that we are comparing HLST to no exercise. Additionally, all participants would not have participated in any sort of strength training within at least the past one-month. As it is difficult to power a feasibility trial on the feasibility outcomes, we have powered the trial on the first indicator of efficacy, which is muscular strength. Our study is approaching all HNCS in the Edmonton and surrounding area. Therefore, there is no viable secondary plan if the target accrual is not reached. Based on an initial review by the Alberta Cancer Registry, there are approximately 1,495 HNCS living in the Edmonton area. Based on our previous studies [13,14,20–22,25,28,52,53], we expect around 50% to be eligible and a recruitment rate of about 6%, which will allow us to reach our target.

## Data collection and management

All data will be directly entered on REDCap, or recorded on case report forms (CRFs) and stored anonymously in a locked cabinet at the Exercise Oncology Research Laboratory, at the University of Alberta before, during, and after the trial. The investigators will provide access to the data file on reasonable request. The investigator is ultimately responsible for the collection and timely reporting of all applicable data entered in CRFs and ensuring they are accurate, original, attributable, complete, legible, contemporaneous, and available when required. A Data Monitoring Committee will not be involved in this study as it is a short-term in its duration, in its early stages, and based on the intervention and outcome of interest, the potential for harm to study participants is deemed to be minimal, if any. De-identified research data will be made publicly available when the study is completed and published.

## Statistical considerations

All randomized participants will be included in the analysis using an intention-to-treat approach. There will be no interim analyses. If missing data is < 10% we will conduct a complete case analysis. If missing data is > 10%, we will employ a multiple imputation missing data strategy [54,55]. Continuous variables will be described using mean (standard deviation) or median (interquartile range). Categorical variables will be described using frequencies (percentages) and confidence intervals. Descriptive analyses will be performed for participant characteristics, feasibility outcomes, adherence, and adverse events. Analysis of covariance will be performed to compare post-intervention between-group differences post-intervention in health-related fitness and patient-reported outcomes for patients with evaluable data. We will adjust for covariates such as baseline physical function levels, cancer stage, HPV status, treatments received (surgery, chemotherapy/radiotherapy, surgery + chemoradiotherapy), and time since treatment completion in the primary analysis. IBM SPSS Statistics version 28 will be used for all statistical analyses. The level of statistical significance will be set at 0.05, and all hypotheses' tests will be two-sided.

Strategies to minimize dropouts and protocol deviations include: 1) a progression period of at least 5-weeks is in place to reach heavy loads (80% to 100% of maximal perceived exertion) in the barbell squat, bench press, and deadlift; 2) reducing the load, number of sets, and/or repetitions; 3) allowing for multiple rest days in between the two weekly exercise sessions; and 4) participants in the Usual Care Group will be offered a 4-week introduction to HLST program and/or referred to a community-based exercise program to encourage their adherence to the protocol, and to thank them for their time in the study. All participants who withdraw from the study will be invited to complete the follow-up assessments to avoid missing data for the intention-to-treat analyses.

## Patient and public involvement

A patient and public involvement panel were not specifically conducted to inform the research question, study design, recruitment or dissemination plan for this study. However, the study coordinator, SMN, is a multiple-time head and neck cancer survivor and used her medical and weight training experiences, along with available evidence to inform this study.

## Ethics and dissemination

The study was approved by the Health Research Ethics Board of Alberta-Cancer Committee (HREBA.CC-24–0021); Trial Registration Number: NCT06289049. All patients will provide written informed consent prior to the beginning of the study. Protocol amendments will be sent to HREBA for review, and study participants will be informed in-person and/or via email immediately, and provided with a revised consent form. The outcomes of the LIFTING 2 trial will be disseminated through peer-reviewed academic journals, conferences, via the webpage www.lifting2study.ca and monthly email updates. Study findings will be published in a journal, presented at scientific conferences and public channels.

## Discussion

Light-to-moderate intensity exercise has proven to be effective for patients with HNC [14,16,17,20,22–25]. Patients with HNC are typically older at diagnosis and present with multiple comorbidities such as hypertension, hyperlipidemia, chronic obstructive pulmonary disease and diabetes; and may develop treatment-related comorbidities such as pneumonia, dysphagia, malnutrition, and dental issues [56]. This may make it challenging to implement exercise compared to other younger cancer groups and those with fewer treatment-related side effects. Feasibility studies are important to understand if further investigation is necessary, considering the sustainability and strength of the research and relevance of the findings [57]. The primary focus of the LIFTING 2 trial will be to further establish the feasibility of HLST in HNCS.

To date, only three studies have examined the effects of a heavy load lifting program in any cancer population; two studies with machine-based interventions in patients with breast cancer receiving or scheduled to receive adjuvant therapy [29,30], and the preceding trial of the proposed LIFTING 2 trial, with a free weight intervention in HNCS post-treatment [28]. The two studies in patients with breast cancer demonstrated improvements in muscular strength, walking economy, increased time to exhaustion during incremental walking, and reductions in lymphedema in the heavy load exercise groups compare to no exercise or low load strength training [29,30]. However, these studies were in patients receiving or scheduled to receive treatment, and were prescribed machine-based exercises. A third study examined the effects of heavy-load strength training on muscle strength, body composition, muscle fibre size, satellite cells, and myonuclei during neoadjuvant chemotherapy in women with breast cancer. The authors reported that upper and lower body muscle strength increased more in participants in the strength training group compared to participants in the control group. Both groups reduced fat free mass while increasing fat mass, and no differences were reported in muscle fibre size. Lastly, myonuclei per fibre increased in the control group whole decreasing in the strength training group in type I muscle fibres, where more nuclei translate to a greater capacity for muscle regrowth and strength [58,59]. The final and most applicable study, the LIFTING trial [28], was a small phase I study that preceded the LIFTING 2 trial and assessed the feasibility of HLST in HNCS. The authors reported an excellent median attendance rate of 95.8%, and no adverse events. Meaningful improvements in upper and lower body muscular strength, and global health status/QoL were reported. However, the LIFTING trial had important limitations and warrants further investigation in order to further establish the feasibility of HLST in HNCS [28]. To our knowledge, no randomized controlled trial has been conducted to examine the feasibility, safety, and efficacy of HLST in HNCS post-treatment.

HLST was selected as the exercise intervention for the LIFTING 2 trial for many reasons. Prior to the pilot LIFTING trial [28], which precedes the LIFTING 2 trial, the exercise studies in patients with HNC focused on rehabilitation interventions and light-to-moderate intensity resistance training [14,16,17,20,22–25]. Preliminary data of HLST in HNCS and patients with breast cancer demonstrates that HLST may be feasible, safe, and effective for improving muscle strength and functional outcomes [28–30,58]. In addition, the literature demonstrates that exercising with heavy loads (>75% 3RM) is safe and effective for improving muscular strength, muscular power, muscular endurance, functional outcomes, and glycemic control, while reducing sarcopenia and retaining motor function in both healthy and diseased older populations like those with balance impairments, breast cancer, diabetes, stroke survivors [29,30,60–63]. Overall, the evidence suggests that resistance training is feasible, safe, and may lead to improvements in muscle strength and function, lean body mass, and is received well by patients with HNC with a high adherence rate reported [19,23,28]. Although it is possible to gain muscular strength with sufficient repetitions at low and moderate loads (≤60% 3RM), training with heavy loads is a requisite for optimizing muscular strength potential and possess a strength-related advantage compared to low loads [26,64].

Moreover, current evidence suggests an association between muscle strengthening activities and mortality and cancer risk reduction. Muscle-strengthening activities were associated with a 10% to 17% lower risk of cardiovascular disease, total cancer occurrence, diabetes, cancer-specific mortality, and all-cause mortality independent of aerobic activities among adults [65,66].

Low muscle mass and sarcopenia, the age-related loss of skeletal muscle mass and strength, are prevalent and independent predictors of poor outcomes in patients with cancer [67–69]. While low muscle mass occurs at any age, older adults experience accelerated declines in muscular strength at a rate of approximately 15% per decade between ages 50 and 70 years, and about 1.5% per year thereafter [60]. HNC is particularly a catabolic disease with up to 80% of patients experiencing low muscle mass, leading to significant muscle depletion and functional decline [70]. The effect is even more pronounced in older patients, compounding the challenges of survivorship. Given that muscle loss is associated with reduced survival, greater treatment-related toxicity, and worse recovery, it is essential to prioritize strategies that preserve or enhance muscle mass and strength during HNC survivorship [71]. Resistance training is promising for mitigating muscle loss, improving functional outcomes, and potentially influencing survival. Although, there is still more research to be conducted in the strength training realm with respect to its impact on mortality, cancer risk reduction, and survival.

The LIFTING 2 trial has several limitations including its modest sample size, the absence of long-term follow-up, and the absence of a correlative (biological) component such as gene expression alternations that may be associated with HNC recurrence. This single-centre study is primarily aimed at further establishing the feasibility of HLST, and is underpowered to determine the efficacy of exercise on clinical outcomes of cancer recurrence. In the future, we plan to scale this intervention to include multiple centres. If this current LIFTING 2 study establishes feasibility, safety, and efficacy, we can then estimate how many sites would be needed to properly power a phase III trial. A multi-site study would then allow for a greater sample size, as well as greater validity and generalizability of our study findings. Potential barriers to implementation that may take place in a future multi-site study may include: variations in procedures and practices between sites, communication breakdown and challenges across different time zones, standardization in exercise delivery methods may be challenging with various levels of experience, and possible differences in protocol training conducted for exercise supervisors at each site. The recurrence rate of all HNC subtypes is approximately 50% but may vary depending on cancer stage and HPV status [72]. We will follow-up on health-related fitness and patient-reported outcomes at post-intervention (12-weeks) to determine the direction and magnitude of short-term changes between the groups. We will not follow-up on any long-term cancer outcomes. We will also determine if there are any meaningful changes in the hypothesized direction, although we acknowledge that there is a low likelihood of demonstrating a statistically significant effect. Minimal clinical important differences in upper and lower body muscular strength changes are not clearly reported in the literature, however, standardized effects sizes of 0.50 standard deviations are widely considered clinically important. Other minimally important changes include: 5.0–6.5 kg for handgrip strength [73]; ≥ 2 repetitions for the 30 second sit to stand test [74]; 14.0–30.5 metres in the 6-minute walk test [75]; 2–10° in active shoulder flexion or abduction [76,77]; 5–10 points for quality of life [78]; 6 points for insomnia [79]; and a ≥ 5% weight loss in one-month or ≥10% weight loss in six-months for malnutrition risk assessment [51]. This information will aid in the development of research objectives and the design of larger exercise studies in HNCS.

The LIFTING 2 trial has several strengths including the understudied patient population of HNCS, the randomized controlled trial design with a non-exercise comparison group, the HLST supervised exercise sessions, the free weight HLST intervention, the validated measurement tools and questionnaires used, and the list of comprehensive outcomes that will be assessed. To our knowledge, the LIFTING 2 trial is the first randomized controlled trial to test the feasibility and safety of a HLST program in HNCS. The study will further establish the feasibility of HLST in HNCS by reporting adherence rates, recruitment rates, retention rates, and adverse events related to the exercise intervention. In addition, the LIFTING 2 trial will also provide preliminary evidence on whether exercise may improve health-related fitness and patient-reported outcomes in HNCS at least one-year post-treatment. This study may inform larger phase II and III trials designed to test the efficacy of strength training on important clinical outcomes including muscle strength, mass, and function, symptom management, and QoL.

## Supporting information

**S1 File. Spirit checklist.**
(PDF)

## Author contributions

**Conceptualization:** Stephanie M Ntoukas, Margaret L. McNeely, Carla M. Prado, Kerry S. Courneya.

**Methodology:** Stephanie M Ntoukas, Margaret L. McNeely, Carla M. Prado, Kerry S. Courneya.

**Writing – original draft:** Stephanie M Ntoukas, Kerry S. Courneya.

**Writing – review & editing:** Stephanie M Ntoukas, Margaret L. McNeely, Carla M. Prado, Kerry S. Courneya.

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
