## [Decision Letter · Decision Letter 0]

PONE-D-25-15890Feasibility and Preliminary Efficacy of Heavy Lifting Strength Training versus Usual Care in Head and Neck Cancer Survivors (the LIFTING 2 Trial): a study protocol for a single-centre, phase II, randomized controlled trialPLOS ONE

Dear Dr. Ntoukas,

Thank you for submitting your manuscript to PLOS ONE. After careful consideration, we feel that it has merit but does not fully meet PLOS ONE’s publication criteria as it currently stands. Therefore, we invite you to submit a revised version of the manuscript that addresses the points raised during the review process.

**ACADEMIC EDITOR: **

The reviewer(s) have recommended revisions to your manuscript.  Therefore, I invite you to respond to the reviewer(s)' comments and revise your manuscript.  Revision of your work does not guarantee acceptance and the resubmission will undergo further peer review. 

We look forward to receiving your revised manuscript.

Kind regards,

Ana Paula Drummond Lage

Academic Editor

PLOS ONE

“Dr. Kerry Courneya will be fully funding this trial out of his Canadian Institutes of Health Research Foundation Grant and his Canada Research Chair funding. This will cover costs of study personnel including casual employees and research assistantships for graduate students. It also includes incidentals such as printing letters and study brochures to mail out for recruitment, cleaning supplies, and transportation to and from our facility. Participants will be reimbursed for all study-related expenses. We have 3 pre-paid parking spots at the University of Alberta which are reserved for participants. If unavailable, they will be reimbursed for all parking costs. CMP was partially funded through Canada’s Research Chair program.”

Reviewers' comments:

Reviewer's Responses to Questions

**Comments to the Author**

1. Does the manuscript provide a valid rationale for the proposed study, with clearly identified and justified research questions?

Reviewer #1: Yes

Reviewer #2: Yes

2. Is the protocol technically sound and planned in a manner that will lead to a meaningful outcome and allow testing the stated hypotheses?

Reviewer #1: Yes

Reviewer #2: Partly

3. Is the methodology feasible and described in sufficient detail to allow the work to be replicable?

Reviewer #1: Yes

Reviewer #2: No

4. Have the authors described where all data underlying the findings will be made available when the study is complete?

Reviewer #1: Yes

Reviewer #2: Yes

5. Is the manuscript presented in an intelligible fashion and written in standard English?

Reviewer #1: Yes

Reviewer #2: Yes

6. Review Comments to the Author

You may also provide optional suggestions and comments to authors that they might find helpful in planning their study.

Reviewer #1: Thank you for the opportunity to review this manuscript titled "Feasibility and Preliminary Efficacy of Heavy Lifting Strength Training versus Usual Care in Head and Neck Cancer Survivors (the LIFTING 2 Trial): a study protocol for a

single-centre, phase II, randomized controlled trial." This is a very strong protocol paper that clearly outlines the rationale, objectives, and methodology of the project. Feasibility studies such as this are extremely important to establishing strong evidence-bases and unfortunately are largely underutilized. I look forward to reading about the results of this study and how the intervention evolves throughout the research process. I have two main suggestions for improvement:

1. Your stated primary outcome is feasibility though you are powering on the efficacy outcome. This is common in implementation trials as well where we cannot power on the implementation outcome, therefore we power on the intervention outcome. It would be good to justify this choice.

2. I suggest adding plans for handling missing data in the analysis to the statistical considerations section.

There are very minor grammatical/spelling errors in a few places. I would recommend a read-through of the entire manuscript and possibly use of grammar-checking software to help clean these minor errors.

Reviewer #2: The LIFTING 2 Trial presents a relevant and timely study evaluating the feasibility and preliminary efficacy of heavy lifting strength training (HLST) in head and neck cancer survivors. The randomized controlled design, inclusion of objective and patient-reported outcomes, and structured intervention protocol are commendable. From a methodological and statistical point of view, I have the following comments:

1. The sample size calculation is based on detecting a large effect size (Cohen’s d = 0.80) with 80% power and α = 0.05, resulting in 38 evaluable patients (19 per group) after accounting for 20% attrition. While an effect size of 0.80 may be supported by prior pilot work (e.g., Ntoukas et al., 2023) and other exercise oncology literature, it remains optimistic. Can the authors elaborate on the rationale for selecting ES = 0.80 as the target? Would the authors consider including a sensitivity analysis to explore power implications for more modest effect sizes (e.g., ES = 0.50), which may be clinically relevant in this population? This would contextualize the feasibility of detecting smaller but still clinically meaningful effects in future trials.

2. The protocol notes an intention-to-treat analysis, but does not specify the approach for handling missing data due to attrition. How do the authors plan to handle missing outcome data if the attrition rate exceeds 20%? Will multiple imputation, mixed-effects modeling, or another method be applied to mitigate bias?

3. The trial uses usual care (UC) as the control arm, which is methodologically acceptable for early-phase feasibility trials. Did the study team consider using light-to-moderate strength training (LMST) as a comparison group instead of UC? Given that LMST has previously shown benefits in this population, would this not serve as a more informative comparator for assessing the incremental benefit of HLST? A comparison against LMST might help determine whether HLST confers superior benefits, while also controlling for placebo and behavioral engagement effects.

4. This is a single-center trial conducted at the University of Alberta, which can streamline recruitment and intervention fidelity. However, it may limit the external validity and generalizability of the findings. Do the authors foresee challenges in generalizing results from a single site? Are there plans to scale this intervention to a multicenter context in future trials, and what barriers to implementation have been considered? A brief discussion of site-specific factors (e.g., infrastructure, clinical culture) that could influence reproducibility elsewhere would be useful.

5. The trial uses stratified randomization based on sex and cancer stage (I–III vs. IV), a reasonable approach for ensuring balance in known prognostic categories. However, HNCS is a clinically heterogeneous group. Could the authors clarify whether any additional prognostic factors were considered during the design phase for inclusion in the randomization scheme or as covariates in the primary analysis? For instance, time since treatment completion, HPV status, baseline physical function, or dysphagia severity may influence strength or QoL outcomes.

6. Due to the nature of the intervention, blinding of participants and investigators is not feasible. However, outcome assessor blinding (especially for strength testing) may still be possible and should be considered to reduce detection bias. Additionally, the use of objective measures alongside self-reported outcomes helps mitigate potential performance bias.

7. PLOS authors have the option to publish the peer review history of their article (what does this mean? ). If published, this will include your full peer review and any attached files.

**Do you want your identity to be public for this peer review?** For information about this choice, including consent withdrawal, please see our Privacy Policy .

Reviewer #1: **Yes: ** Mary-Louise E. Millett

Reviewer #2: No

---

## [Author Response · Author response to Decision Letter 1]

20 May 2025

Hello,

A detailed response to all editor comments can be found in the attached file entitled 'Response to Reviewers.' Please do not hesitate to reach out if anything else is required. Thank you.

---

## [Decision Letter · Decision Letter 1]

Feasibility and Preliminary Efficacy of Heavy Lifting Strength Training versus Usual Care in Head and Neck Cancer Survivors (the LIFTING 2 Trial): a study protocol for a single-centre, phase II, randomized controlled trial

PONE-D-25-15890R1

Dear Dr. Ntoukas,

We’re pleased to inform you that your manuscript has been judged scientifically suitable for publication and will be formally accepted for publication once it meets all outstanding technical requirements.

Kind regards,

Ana Paula Drummond Lage

Academic Editor

PLOS ONE

Additional Editor Comments (optional):

Thank you for submitting the revised version of your manuscript entitled "Feasibility and Preliminary Efficacy of Heavy Lifting Strength Training versus Usual Care in Head and Neck Cancer Survivors (the LIFTING 2 Trial): a study protocol for a single-centre, phase II, randomized controlled trial" (PONE-D-25-15890R1).

Both reviewers appreciated the thoughtful responses to the comments and has no further recommendations. While the format of the revision responses could have been more intuitive, all concerns were adequately addressed.

I am pleased to inform you that your manuscript has been accepted for publication in PLOS ONE.

Congratulations, and thank you for choosing PLOS ONE to share your work.

Reviewers' comments:

Reviewer's Responses to Questions

**Comments to the Author**

1. Does the manuscript provide a valid rationale for the proposed study, with clearly identified and justified research questions?

Reviewer #1: Yes

Reviewer #2: Yes

2. Is the protocol technically sound and planned in a manner that will lead to a meaningful outcome and allow testing the stated hypotheses?

Reviewer #1: Yes

Reviewer #2: Yes

3. Is the methodology feasible and described in sufficient detail to allow the work to be replicable?

Reviewer #1: Yes

Reviewer #2: Yes

4. Have the authors described where all data underlying the findings will be made available when the study is complete?

Reviewer #1: Yes

Reviewer #2: Yes

5. Is the manuscript presented in an intelligible fashion and written in standard English?

Reviewer #1: Yes

Reviewer #2: Yes

6. Review Comments to the Author

You may also provide optional suggestions and comments to authors that they might find helpful in planning their study.

Reviewer #1: I thank the authors for thoughtfully addressing my comments. I have no further recommendations.

Reviewer #2: Thanks for addressing the comments and revising the manuscript.

An area for Improvement that might be useful when planning/conducting the study:

(Comment 1): Add a sensitivity analysis or caveat about effect size optimism.

7. PLOS authors have the option to publish the peer review history of their article (what does this mean? ). If published, this will include your full peer review and any attached files.

**Do you want your identity to be public for this peer review?** For information about this choice, including consent withdrawal, please see our Privacy Policy .

Reviewer #1: **Yes: ** Mary-Louise E. Millett

Reviewer #2: No

---

## [Editor Report · Acceptance letter]

PONE-D-25-15890R1

PLOS ONE

Dear Dr. Ntoukas,

I'm pleased to inform you that your manuscript has been deemed suitable for publication in PLOS ONE. Congratulations! Your manuscript is now being handed over to our production team.

Kind regards,

on behalf of

Dr. Ana Paula Drummond Lage

Academic Editor

PLOS ONE